# Instruction-Guided Visual Masking

**Jinliang Zheng**[*1,2], **Jianxiong Li**[*1], **Sijie Cheng**[1], **Yinan Zheng**[1],
**Jiaming Li**[1], **Jihao Liu**[3,2], **Yu Liu**[2], **Jingjing Liu**[†1], **Xianyuan Zhan**[†1,4]
[1] AIR, Tsinghua University, [2] Sensetime Research
[3] MMLab, CUHK, [4] Shanghai AI Lab
{zhengjl23, li-jx21}@mails.tsinghua.edu.cn
zhanxianyuan@mail.tsinghua.edu.cn

## Abstract

Instruction following is crucial in contemporary LLM. However, when extended to multimodal setting, it often suffers from misalignment between specific textual instruction and targeted local region of an image. To achieve more accurate and nuanced multimodal instruction following, we introduce *Instruction-guided Visual Masking* (IVM), a new versatile visual grounding model that is compatible with diverse multimodal models, such as LMM and robot model. By constructing visual masks for instruction-irrelevant regions, IVM-enhanced multimodal models can effectively focus on task-relevant image regions to better align with complex instructions. Specifically, we design a visual masking data generation pipeline and create an IVM-Mix-1M dataset with 1 million image-instruction pairs. We further introduce a new learning technique, *Discriminator Weighted Supervised Learning* (DWSL) for preferential IVM training that prioritizes high-quality data samples. Experimental results on generic multimodal tasks such as VQA and embodied robotic control demonstrate the versatility of IVM, which as a plug-and-play tool, significantly boosts the performance of diverse multimodal models, yielding new state-of-the-art results across challenging multimodal benchmarks. Code, model and data are available at https://github.com/2toinf/IVM.

## 1 Introduction

Multimodal instruction following is a fundamental multimodal task, powering a wide-range of applications such as visual question answering (VQA) [18], visual captioning [1, 41], and embodied robotic control [14]. To effectively solve this task, one critical capability required is nuanced image-language grounding, which current multimodal models grow implicitly and slowly through data-intensive end-to-end training without explicit grounding supervisions. Two challenges emerge in this indirect learning of image-instruction alignment: 1) How to accurately localize targeted image regions that corresponds to a specific textual instruction, as illustrated in Figure 1. 2) How to generalize to diverse visual representations (*e.g.*, same object with different colors, compositions, or backgrounds) that reflect similar textual instruction (e.g., Q3 in Figure 1). Lacking an effective and direct solution to these challenges, the most advanced Large Multimodal Models (LMMs) [1, 6, 41, 14] still suffer from hallucinations even when trained with high-quality data in the magnitude of billions [34].

We introduce *Instruction-guided Visual Masking* (IVM), a versatile plug-and-play model designed to enhance multimodal instruction following via nuanced surgical visual grounding. To eliminate the distraction of instruction-irrelevant visual regions, IVM automatically masks out these regions to sharpen the focus of instruction following, and meticulously crops visual input to tailor for a specific

---

[*]Equal contribution
[†]Corresponding author

38th Conference on Neural Information Processing Systems (NeurIPS 2024).

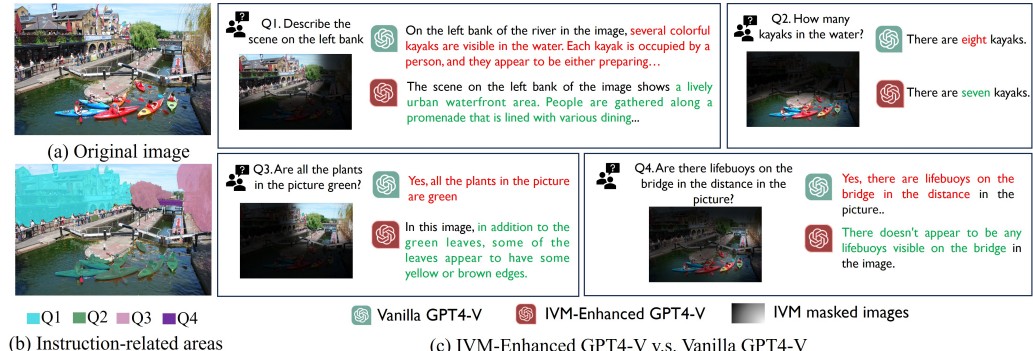

Figure 1: The most advanced LMMs (e.g. GPT4-V) still fail on complex instruction following tasks. With IVM assistance to simplify visual inputs, existing LMMs can gain significant improvement.

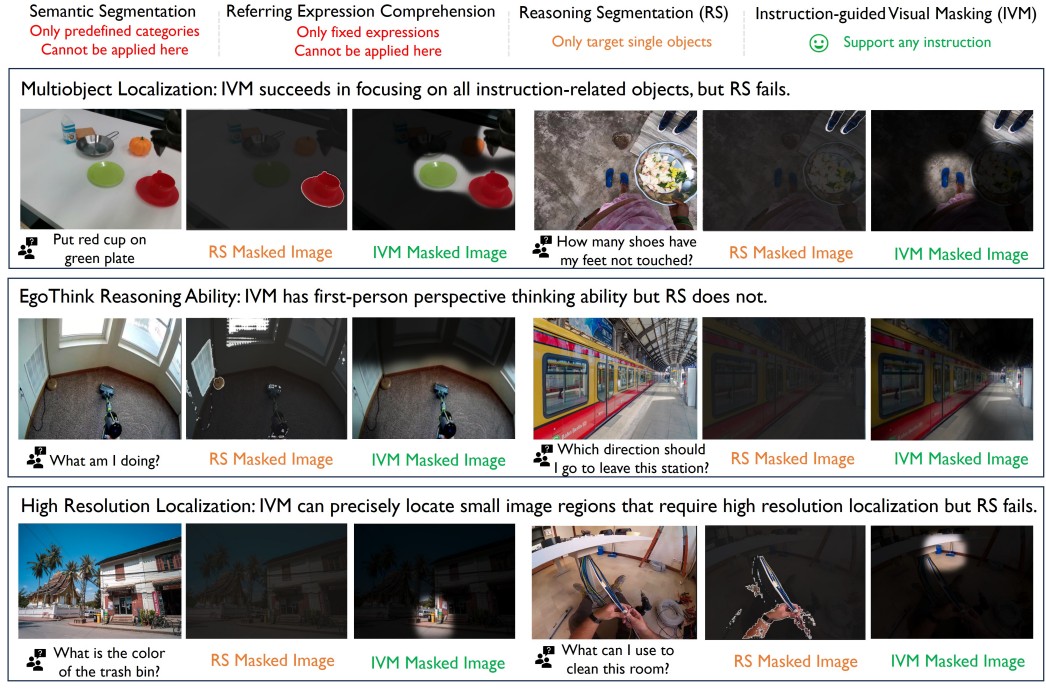

Figure 2: Comparison between IVM and Reasoning Segmentation (RS) [31]. Traditional methods such as semantic segmentation [68] and referring expression comprehension [64] are limited to fixed categories or fixed instruction formats, thus inapplicable to complex instruction following tasks. RS has reasoning ability, but only allows single object localization. IVM, instead, is universally applicable to any instruction.

instruction and enforce multimodal models to zoom in on task-related visual content. Existing visual grounding methods are limited either to predefined object categories, which cannot cover diverse instruction-related visual content; or they subscribe to a fixed instruction format, which restricts the expressiveness of instructions. As shown in Figure 2, such simplistic grounding techniques often fail to comprehend complex instruction-following tasks.

Learning an IVM model requires pixel-level, fine-grained, instruction-guided mask annotations that provide explicit grounding supervisions. To create such a dataset, we build a LLM-empowered Mixture of Expert pipeline with SOTA visual grounding models [52, 50, 31, 20] to efficiently create abundant reliable labels. To compensate the noises in auto-generated labels, we further manually label a smaller dataset with clean annotations, and integrate the two into an IVM-Mix-1M dataset that contains 1 million image-instruction pairs.

To reduce demand on costly human labels and ensure optimized utility of machine-generated labels, we employ a Discriminator-Weighted Supervised Learning (DWSL) framework for IVM training, inspired by recent advances in offline imitation learning [60]. Specifically, we introduce a discriminator to assign weights to masks, where high values are assigned to high-quality annotations and vice versa. Thus, these weights generated by the discriminator can naturally act as a weighting function for the IVM training objective, allowing for a preferential training process that prioritizes learning from reliable samples and discards misleading ones.

Extensive experiments demonstrate great versatility of the IVM model when integrated into existing multimodal chatbots (commercial and open-sourced) without fine-tuning. Our IVM-enhanced LMMs gain significant performance improvement across new challenging benchmarks such as V*Bench [58], EgoThink [10] and POPE [34], achieving new state of the art. IVM model also proves valuable in vision-language robotic manipulation tasks, where data collection is notoriously challenging and generalization is a major concern [35]. With the integration of IVM, our enhanced robot model exhibits boosted performance and better generalization capabilities.

Our contributions are summarized as follows: 1) We propose Instruction-guided Visual Masking (IVM), a novel approach that serves as a versatile plug-and-play module to enhance multimodal models through visual grounding. 2) We introduce the IVM-Mix-1M dataset and propose an LLM-empowered Mixture of Expert pipeline to create visual grounding labels. 3) We present the DWSL algorithm for IVM training that automatically prioritizes high-quality training samples.

## 2   Related Work

**Large Multimodal Models**.   LLaVA [41] first demonstrates promising capabilities in following complex instructions. Subsequent works such as LLaVA-1.5 [38], MiniGPT4 [69] Qwen-VL [6] and CogVLM [57], further enhance LMMs via refined model design and enriching the quality of training data, achieving state-of-the-art performance on diverse downstream tasks including visual grounding [36], visual reasoning [55], visual question and answering [18]. Moreover, by integrating the robotics action modality, LMMs perform versatile planning and manipulation in instruction-driven robotics tasks. Notable studies in this line of inquiry include PaLM-E [14], the series of RT models [8, 9, 54], and text-guided video planning diffusion models [15, 62, 7]. Despite the success, LMMs still struggle with complex visual grounding challenges, often misreading instruction-irrelevant visual contents (Figure 1). To address this, researchers have tried to adapt existing visual modules to higher-resolution images to obtain better perception [40], but with limited improvement.

**Visual Grounding Tasks**.   Visual grounding requires precisely localizing image regions corresponding to a referring expression, among which the RefCOCO series [64] is the most well-known benchmark, and numerous public visual grounding data are available [36, 63, 19]. Recently, LMMs incorporate these visual grounding data via visual-instruction tuning [41, 38, 69], establishing new SOTA in this area [50]. To further broaden the reasoning ability of visual grounding, LISA [31] introduces a new task, reasoning segmentation, which demands higher capabilities in instruction comprehension. However, visual grounding is still limited to align simple instruction with specific objects, which cannot adapt to more complex instruction following tasks (*e.g.* Figure 2).

**Visual Grounding Augmented LMMs**.   Recently, a series of visual grounding methods emerged to enhance the performance of LMMs in complex visual scenes. V* [58] employs a heuristic search strategy to search, locate, and crop image areas relevant to instructions through a multi-step iterative process. VisualCot [50] is trained end-to-end with a customized dataset to achieve target localization capabilities. These two methods allow LMMs to dynamically focus on visual inputs until the correct answer is derived. However, these complex inference pipelines lead to substantial computational overhead, and their heuristic designs further hinder the extension beyond VQA to other multimodal instruction following tasks such as robotic control.

Besides these explicit strategies incorporating additional visual grounding modules, other studies pursue refining data or introducing extra training targets to enhance the grounding capabilities of LMMs implicitly. ViGor [61] proposes a fine-grained reward modeling to enhance visual grounding of LMMs, and SynGround [23] introduces a pragmatic framework for image-text-box synthesis tailored for visual grounding. These methods, however, are primarily focused on the visual grounding task itself, overlooking its influence on downstream multimodal instruction following tasks.

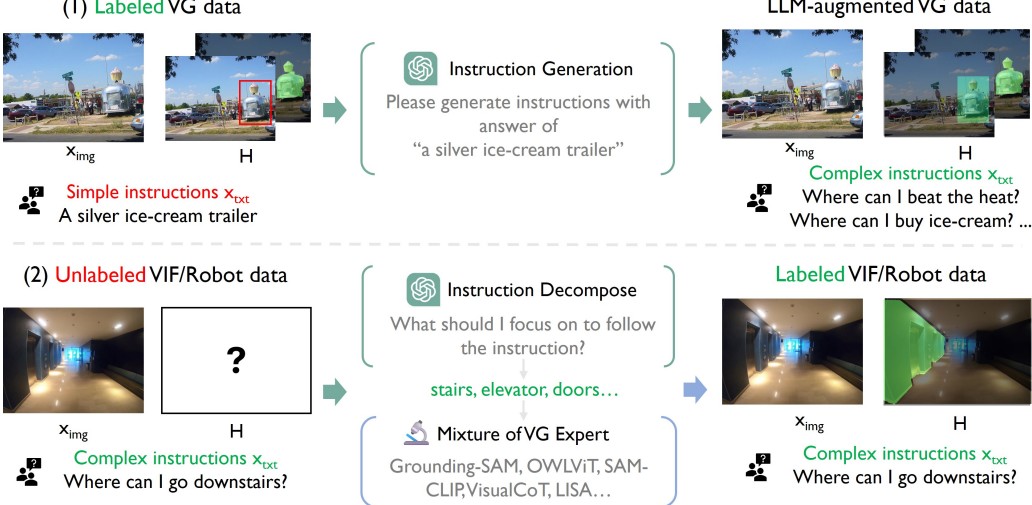

Figure 4: LLM-empowered Mixture-of-Expert pipeline for auto-annotation. (1) For labeled VG data, we utilize an LLM to generate complex instruction annotations. (2) For unlabeled VIF or robot data, we first use an LLM to simplify the instruction and then leverage a mixture of VG models to generate candidate labels.

Distinct from previous efforts, this paper introduces a generic visual grounding model that is adaptable to any multimodal instruction following tasks, and provides a systematic investigation into the advantages of integrating an additional visual grounding model into downstream applications.

# 3 Instruction-Guided Visual Masking

To help multimodal models focus on instruction-sensitive image regions without distractions from irrelevant visual elements, we introduce Instruction-guided Visual Masking (IVM), a versatile plug-and-play model that enhances multimodal instruction following via surgical targeted visual grounding.

## 3.1 Problem Definition

IVM aims to produce a heatmap $\mathbf{H}$, given an image $\mathbf{x}_{\mathrm{img}}$ and a textual instruction $\mathbf{x}_{\mathrm{txt}}$. The heatmap $\mathbf{H}$ identifies the critical image region to follow the instructions, as illustrated in Figure 3, allowing multimodal models to easily zoom in on targeted image regions while ignoring neighboring areas.

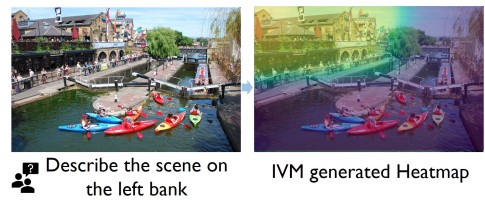

Describe the scene on the left bank     IVM generated Heatmap

Figure 3: Instruction-guided Visual Masking.

This formulation evokes the problem definition of Reasoning Segmentation (RS) [31]. There are two main differences: 1) IVM addresses a more challenging problem. RS tries to target single objects from simple instructions, *e.g.*, "what is.., where is.., who is...", while IVM aims to include all instruction-related visual regions within the image given any instruction, which demands advanced and nuanced image-language grounding ability (as illustrated in Figure 2). 2) RS has clear ground truths but IVM does not. The instructions in RS primarily correspond to simple and semantic-meaningful objects that are straightforward for human annotations. IVM, however, deals with broader and more ambiguous instruction-related regions (*e.g.*, the left bank regions in Figure 3), making the training and annotating much more challenging.

## 3.2 Data Preparation

To train an IVM model, the first main challenge is the scarcity of training data. Most existing Visual Grounding (VG) datasets [64, 31] typically feature simple instructions focused primarily on prominent objects within images, lacking both diversity and complexity required for IVM. To tackle this, we

compiled one million data from various sources, including labeled visual grounding, unlabeled multi-modal instruction following, and robotics data. As outlined in Section 3.1, scaling human annotations is challenging due to the high complexity of such data. Therefore, we introduce an *LLM-empowered Mixture of Expert pipeline* that integrates SOTA visual grounding models to efficiently generate reliable annotations. We further manually annotate a smaller dataset to compensate inaccuracies in auto-generated labels. The resulted combined dataset, IVM-Mix-1M, comprises one million data samples ready for IVM training, which can be found in `https://github.com/2toinf/IVM`.

**LLM-empowered Mixture of Expert Annotation Pipeline.** Leveraging the power of LLM, this pipeline can efficiently generate high-quality annotation, which consists of two components (Figure 4): 1) *Labeled visual grounding data.* We collect 250K labeled *VG* data from multiple sources including VG caption [36], Flickr30K [63], VSR [3], OpenImage [30], and RefCoCo [64, 37], which provide bounding boxes with simple instructions for each image. To increase the diversity and complexity of instructions, we utilize GPT-4 [1], known for its robust language understanding and generation capabilities, to create diverse instruction-answer pairs based on existing language instructions. 2) *Unlabeled Visual-Instruction-Following (VIF) and robotics data.* We sample a 700K subset from LLaVA-Instruction-tuning [41] for VQA-type data, and a 50K subset from OpenX [54] for robotics data. Given that these data lack grounded labels but contain complex instructions, we use GPT-4 to simplify the language instructions by prompting it to infer the names of targeted objects necessary for following the instructions. These simplified instructions then guide existing VG models to generate candidate labels. To ensure the quality of these labels and compensate for the ambiguous nature of the IVM task, we integrate proposals from several *VG* experts, such as Grounding-Sam [49], LISA [31], AlphaClip [52], and OwlViT [20], via an ensemble approach.

**Manual Annotation**. Despite integrating the most advanced models, the auto-generation design still faces challenges that can lead to data inaccuracies. First, employing LLM to simplify or complicate language annotations without considering image content can introduce uncontrollable biases. Second, as the task exceeds the capabilities of existing models, it becomes impossible to totally exclude low-quality annotations that contain irrelevant visuals or mistakenly filter out critical contents. Thus, to enhance the overall quality of the dataset, we further manually annotate a 10K subset of the constructed dataset to inject human expert knowledge.

**Data Analysis**. Here, we provide quantitative analysis on the IVM-Mix-1M dataset. Figure 5 depicts the data quantities w.r.t the percentage of annotated instruction-related image area. Here, each ratio range is further categorized by different data sources, where manually annotations are treated as a separate category (Human), while all others are machine-generated. Our analysis reveals that the instruction-related image regions only occupy a small fraction of the total image area (*e.g.* most data have less than 40% instruction-relevant image regions), indicating that most visual contents may cause distraction and corroborating the necessity of visual masking for instruction following tasks.

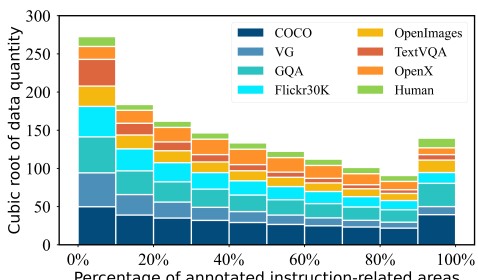

Figure 5: Data analysis on the IVM-Mix-1M dataset: data quantity v.s percentage of instruction-related areas.

### 3.3 Discriminator-Weighted Supervised Learning Framework

The challenge now is to train the IVM model with a small high-quality human-annotated dataset ($\mathcal{D}_e$) as well as a large but mixed-quality auto-generated dataset ($\mathcal{D}_o$). Training naively on the combined dataset may yield suboptimal results due to inaccuracies in auto-generated labels, while solely using limited human-annotated data is insufficient. Inspired by recent advances in imitation learning using mixed-quality data [60, 65], we employ a Discriminator-Weighted Supervised Learning (DWSL) framework to effectively leverage the strengths of both auto- and human-annotated data.

**Discriminator Training**. Specifically, we introduce a discriminator $d$ optimized by Eq. (1) to assign high weights to high-quality annotations and vice versa:

$$\min_d \mathbb{E}_{(\mathbf{x}_{\mathrm{img}}, \mathbf{x}_{\mathrm{txt}}, \mathbf{H}) \sim \mathcal{D}_e} \left[ -\log d(\mathbf{x}_{\mathrm{img}}, \mathbf{x}_{\mathrm{txt}}, \mathbf{H}) \right] + \mathbb{E}_{(\mathbf{x}_{\mathrm{img}}, \mathbf{x}_{\mathrm{txt}}, \mathbf{H}) \sim \mathcal{D}_o} \left[ -\log(1 - d(\mathbf{x}_{\mathrm{img}}, \mathbf{x}_{\mathrm{txt}}, \mathbf{H})) \right],$$
(1)

**Stage I: Discriminator Training**

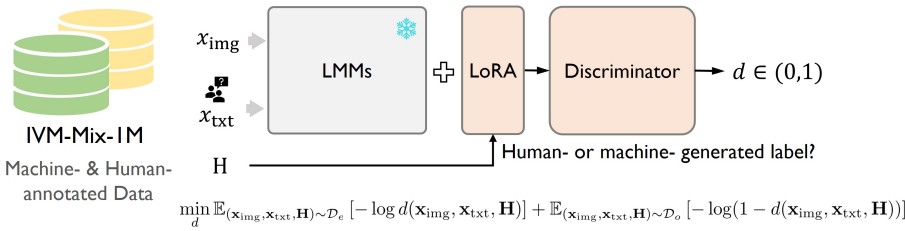

$$\min_d \mathbb{E}_{(\mathbf{x}_{\text{img}}, \mathbf{x}_{\text{txt}}, \mathbf{H}) \sim \mathcal{D}_e} \left[ - \log d(\mathbf{x}_{\text{img}}, \mathbf{x}_{\text{txt}}, \mathbf{H}) \right] + \mathbb{E}_{(\mathbf{x}_{\text{img}}, \mathbf{x}_{\text{txt}}, \mathbf{H}) \sim \mathcal{D}_o} \left[ - \log(1 - d(\mathbf{x}_{\text{img}}, \mathbf{x}_{\text{txt}}, \mathbf{H})) \right]$$

**Stage II: Discriminator Weighted Supervised Learning**

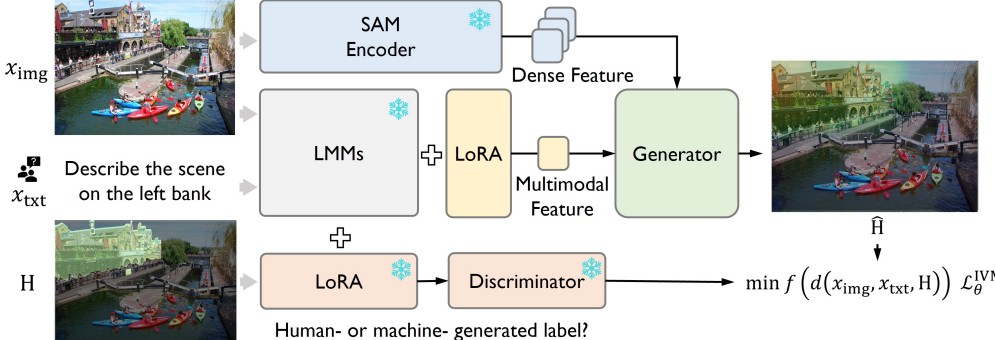

Figure 6: **IVM model architecture and training pipeline.** Stage I: A LoRA-tuned LMMs is trained to discriminate human- and machine-annotated data. Stage II: A frozen SAM vision backbone and a LoRA-tuned LMMs are utilized to extract dense image features and multimodal representations, respectively. These features are then fed into a generator for dense prediction and is trained via DWSL. Same color represents the same model. See Appendix C.1 for more details.

where $(\mathbf{x}_{\text{img}}, \mathbf{x}_{\text{txt}}, \mathbf{H})$ are image-instruction-heatmap pairs sampled from $\mathcal{D}_o$ and $\mathcal{D}_e$ datasets. Eq. (1) is similar to the one in GAN [17], but the "fake" data in [17] is replaced by machine-generated data from $\mathcal{D}_o$. After training with Eq. (1), the discriminator $d$ assigns high weights to high-quality human-annotated data from $\mathcal{D}_e$ and relatively high values to similarly high-quality data from $\mathcal{D}_o$ that aligns with human preferences, acting as a judge for annotation quality.

**Discriminator-weighted IVM Training**. Then, we apply the trained discriminator as a weighting function for the IVM training objective:

$$\min_\theta \mathbb{E}_{(\mathbf{x}_{\text{img}}, \mathbf{x}_{\text{txt}}, \mathbf{H}) \sim \mathcal{D}_o \cup \mathcal{D}_e} \left[ f\left( d(\mathbf{x}_{\text{img}}, \mathbf{x}_{\text{txt}}, \mathbf{H}) \right) \mathcal{L}_\theta^{\text{IVM}}(\mathbf{x}_{\text{img}}, \mathbf{x}_{\text{txt}}, \mathbf{H}) \right], \tag{2}$$

$$\mathcal{L}_\theta^{\text{IVM}}(\mathbf{x}_{\text{img}}, \mathbf{x}_{\text{txt}}, \mathbf{H}) = \lambda_{\text{bce}} \mathbf{BCE}(\hat{\mathbf{H}}_\theta, \mathbf{H}) + \lambda_{\text{dice}} \mathbf{DICE}(\hat{\mathbf{H}}_\theta, \mathbf{H}), \tag{3}$$

where $\lambda_{\text{bce}}$ and $\lambda_{\text{dice}}$ are set to 1.0 and 1.0 to balance the binary cross-entropy loss (**BCE**) and the DICE loss for segmentation (**DICE**) [28], respectively. $f(x) \geq 0$ can be any non-negative, non-decreasing function. For simplicity, we set $f(x) := \min(\max(0.1, x), 1)$. This allows the weighting function $f(d(\cdot))$ in Eq. (2) to dynamically prioritize training with high-quality data determined by the discriminator $d$. This approach maximizes the usage of reliable annotations in $\mathcal{D}_o$ to compensate for the small $\mathcal{D}_e$, while minimizing the impact of low-quality data in $\mathcal{D}_o$, thus optimizing performance.

### 3.4 Model Architecture

The overall model framework is illustrated in Figure 6. Due to its complexity, IVM requires both reasoning and precise localization of the target object, closely paralleling reasoning segmentation [31]. Consequently, for the heatmap generator, we adopt a model design similar to that of LISA [31]. Specifically, we first extract dense image features using an isolated vision backbone and multimodal representation from an LMM, which processes image-instruction pairs. These two types of features are then fed into a lightweight generator that integrates them to produce a dense prediction.

For the discriminator, we deploy a lightweight discriminator that encodes the segmentation map using a two-layer convolution network. This discriminator interacts with the outputs of the LMMs through multiple cross-attention operators and finally outputs a quality score for each sample.

**Trainable Parameters**. To enhance training efficiency, we freeze the pre-trained large foundation models and perform LoRA finetuning [25]. The vision backbone, inherited from *Segment Anything Model* [29], is completely frozen, while the lightweight generator and discriminator are fully finetuned. Notably, we utilize a shared LMM for both the generator and discriminator branches but employing separate LoRA parameters to avoid interference between the two tasks.

## 4 Experiments

In this work, we employ *LLaVA-7B* [42] as the LMM and *SAM-H* [29] as the vision backbone for our IVM model (Figure 6), which is trained on the IVM-Mix-1M dataset using the proposed DWSL algorithm. More details on the architecture and training can be found in Appendix C. We conduct extensive experiments to assess the effectiveness of the IVM model. Specifically, we utilize the heatmap generated by the IVM for image post-processing. These processed images can then be seamlessly fed into downstream multimodal models for diverse tasks, as shown in Figure 7. Unless otherwise specified, we use the image post-processing method of overlaying and cropping to discard instruction-irrelevant image content. A detailed discussion on post-processing methods is presented in Section 4.2. We also provide more evaluation results and analysis in Appendix E.

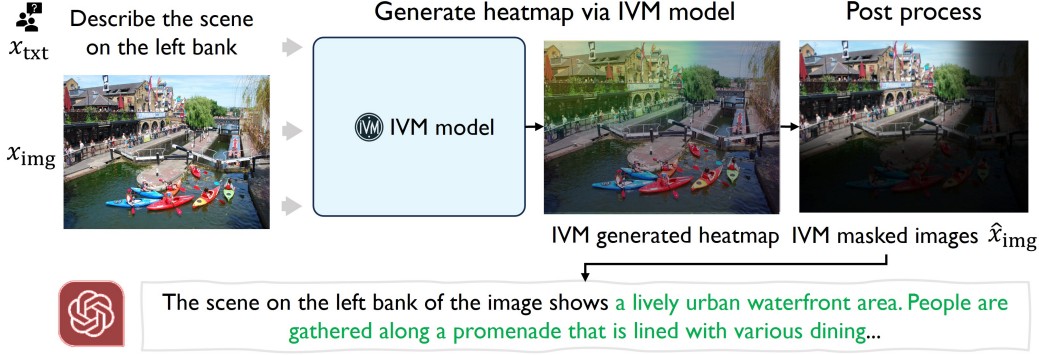

Figure 7: **IVM inference pipeline.** IVM generates heatmap given a pair of image and instruction. Then, instruction-irrelevant visual areas are masked out via post process methods. LMMs can correctly follow the instruction given the masked images.

Table 1: V* bench results.

| LMMs | Attribute(%) | Spatial(%) | Overall(%) |
|---|---|---|---|
| *Open-Sourced LMMs* | | | |
| BLIP2 [33] | 27.0 | 53.9 | 37.7 |
| MiniGPT-4 [69] | 30.4 | 50.0 | 38.2 |
| InstructBLIP [12] | 25.2 | 47.4 | 34.0 |
| Otter [32] | 27.0 | 56.6 | 38.7 |
| LLaVA-1.5 [38] | 43.5 | 56.6 | 48.7 |
| *Commercial Chatbots* | | | |
| Bard [45] | 31.3 | 46.1 | 37.2 |
| Gemini-Pro [53] | 40.9 | 59.2 | 48.2 |
| GPT4-V [1] | 51.3 | 60.5 | 55.0 |
| *Specific Visual Search Models* | | | |
| SEAL [58] | 74.8 (+23.5) | 76.3 (+15.8) | 75.4 (+20.4) |
| IVM-Enhanced GPT4-V | **87.0 (+35.7)** | **72.4 (+11.9)** | **81.2 (+26.2)** |

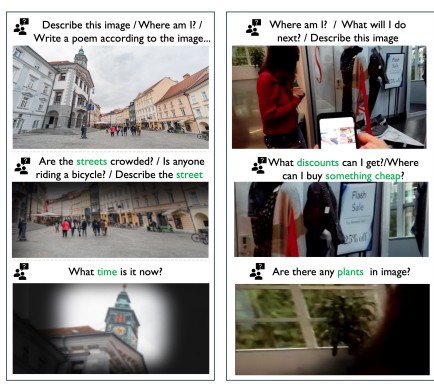

Figure 8: IVM can handle various instructions, ranging from retaining entire images for captioning (row 1) to localizing unique objects (row 2 and 3).

Table 2: Results on other multimodal benchmarks. MME* denotes the aggregate of scores from -p and -c.

| LMMs | #Param | EgoThink | POPE | MME* | GQA | SQA | VQAv2 |
|---|---|---|---|---|---|---|---|
| InstructBLIP [12] | 13B | - | 78.9 | 1212.8 | 49.5 | 60.5 | - |
| Qwen-7B [6] | 7B | - | - | - | 58.3 | 67.1 | 78.8 |
| SEAL-7B [58] | 7B | - | 82.4 | 1129 | - | - | - |
| LLaVA-7B [38] | 7B | 51.1 | 85.9 | 1748 | 62.0 | 70.2 | 78.5 |
| LLaVA-13B [38] | 13B | 55.2 | 85.9 | 1834 | 67.1 | 71.6 | 80.0 |
| LISA [31]-Enhanced LLaVA-7B | 20B | 47.9 (-3.2) | 80.0 (-5.9) | 1560 (-188) | 56.6 (-5.4) | 69.3 (-0.9) | 78.2 (-0.3) |
| IVM-Enhanced LLaVA-7B | 14B | **54.5 (+3.4)** | **87.2 (+1.3)** | **1806 (+58)** | **62.2 (+0.2)** | **70.2 (-)** | **79.0 (+0.5)** |

## 4.1 Main Results

**Integration with Commercial Chatbot**. We use GPT4-V [1] as the base model. Considering the superior perception and reasoning capability of GPT4-V, we evaluate IVM-enhanced GPT4-V on V*bench [58], a recently proposed challenging VQA-type benchmark characterized by images with abundant redundancies. Results are presented in Table 1. The accuracy of the vanilla GPT4-V is mediocre (55.0%). Our IVM model, however, can significantly improve the performance (+26.2%) and establish a new state of the art on this benchmark, even surpassing the task-specialized SEAL [58] that requires a complex heuristic visual search pipeline.

**Integration with Open-sourced LMMs**. To demonstrate the versatility of our IVM model, we further integrate it into an open-sourced LMM, LLaVA-7B [38]. We conduct extensive experiments across various benchmarks, including EgoThink [10], POPE [34], MME [16], GQA [27], SQA [44], and VQAv2 [18]. As shown in Table 2, our IVM-enhanced LLava-7B gains consistent performance improvements, achieving comparable performance to (even surpassing) LLaVA-13B on EgoThink, POPE and MME. Although IVM-enhanced LLaVA-7B and LLaVA-13B [38] have roughly the same number of parameters, the latter integrates more powerful pretrained foundation models. In contrast, our IVM model allows the 7B model to outperform the 13B model by merely simplifying visual input, further validating the power of visual masking.

Meanwhile, IVM-enhanced LLaVA-7B does not show significant gains on GQA, SQA and VQAv2, which is expected, as these benchmarks do not heavily rely on grounding capabilities: VQAv2 and GQA contain relatively simple visual input where most regions of the images are instruction-relevant, while SQA primarily focuses on assessing model reasoning capability.

**Comparison with Reasoning Segmentation Model**. We also compare against LISA [31], which is most analogous to IVM. We provide carefully tailored prompts like *"what should we focus on the image to follow the given instruction? Give me the seg"* to extend LISA into visual masking task. However, even with larger 13B model and extensive tuning of input prompt, masks generated by LISA consistently result in severe performance degradation on all tasks.

**Evaluation on Real Robotic Control**. We also plug the IVM model into robot control tasks to help robot model improve generalization. Specifically, we evaluate a language-conditioned behavior cloning (LCBC) robot agent trained with or without IVM masked images. Figure 9 clearly demonstrates that without IVM assistance, the LCBC robot agent suffers from severe performance drop when noticeable distractions are applied. With IVM assistance, however, the agent consistently pays close attention to correct instruction-related image regions, performing robustly against diverse distractions such as human disturbances and numerous task-irrelevant objects of various colors and shapes. This demonstrates promising potentials of using IVM to enhance embodied agents to follow complex instructions in unseen scenes with plenty distractions.

## 4.2 Ablation

We ablate the key components of IVM and report overall accuracy improvement(%) of IVM-Enhanced GPT4-V, evaluated on V* bench [58] due to its high demand on precise visual grounding abilities.

**Training Data**. We investigate the impact of IVM-Mix data characteristics on IVM performance from two key perspectives: 1) Large machine-annotated data volume clearly enhances IVM model performance, as illustrated by the progressive improvement in Figure 10 (a) with increased machine-annotated data volume (red, blue and yellow line). This demonstrates the effectiveness of our proposed LLM-empowered Mixture-of-Expert pipeline in generating reliable data for IVM training. 2) Figure 10 (a) also reveals that incorporating human annotations significantly boosts training

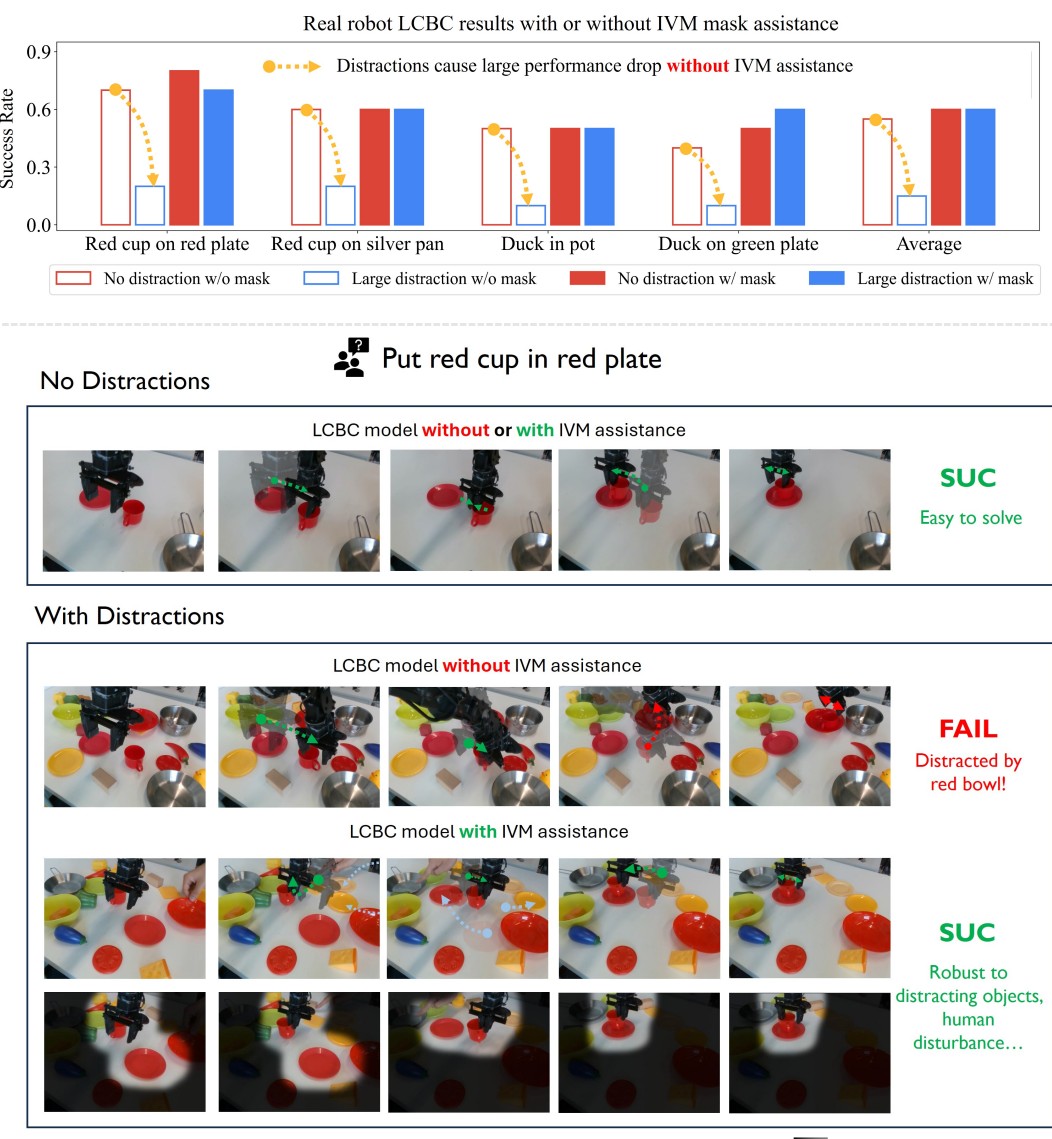

Figure 9: Real robot results with or without IVM assistance. IVM greatly helps LCBC agent to overcome major distractions, enjoying better robustness and generalization. See Appendix C.4 for experiment setups.

efficiency (red and blue *v.s* yellow line), highlighting the critical role of introducing human preferences in IVM-Mix-1M dataset, despite its relatively small volume compared to machine-annotated data (only 1:100).

**DWSL Framework**. We also explore the efficacy of the DWSL framework in Figure 10 (a) by comparing IVM training using: 1) DWSL (red line), 2) traditional Supervised Learning (SL) without DWSL (blue line), and 3) SL on limited human data (gray line). The results demonstrate that DWSL effectively leverages both human- and auto-annotated data, particularly as the volume of machine-annotated data increases, enjoying higher asymptotic performances. This is expected as machine-annotated data often contain inaccuracies and training naively using all these data can lead to suboptimal results. Meanwhile, the limited human data alone cannot provide satisfactory outcomes. DWSL, however, addresses these challenges by dynamically prioritizing good samples and discarding misleading ones, resulting in stable and improved results. This is further illustrated in Figure 10 (b) which visualizes the outputs of the discriminator for each sample, where the discriminator can correctly retain good samples (e.g. Human) and filter out low-quality data with lower weights.

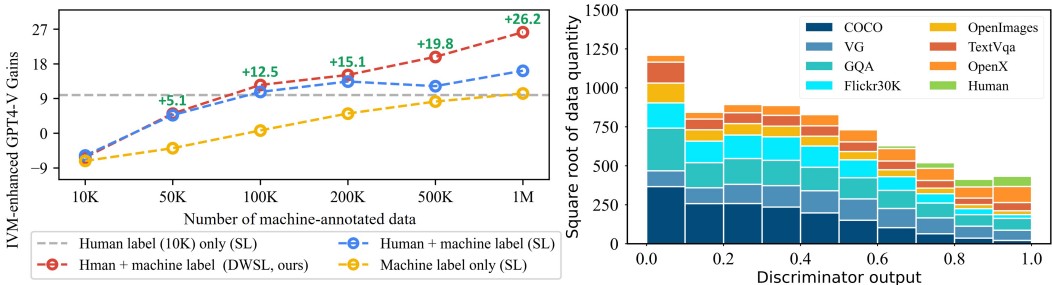

(a) IVM-enhanced GPT-4V gains trained with different data and methods

(b) Discriminator output values for different data sources

Figure 10: Ablations on training data and the proposed DWSL framework.

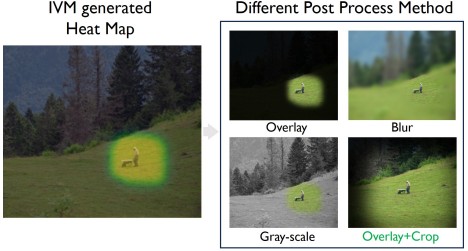

Figure 11: Different mask deployment methods.

Table 3: Ablations on different mask deployment methods on the V*bench.

|  | Overlay | Blur | Gray-scale |
|---|---|---|---|
| w/ crop | **+26.2** | +24.4 | +22.1 |
| w/o | +19.1 | +17.2 | +10.2 |

**Mask Deployment Strategy**. We investigate the impact of mask deployment strategy on downstream applications. While more complex solutions such as visual search algorithms [58] can be employed, our investigation focuses solely on simpler approaches to understand the intrinsic capabilities of IVM model. Specifically, we examine four basic masking methods: overlay, blur, grayscale, and cropping, as illustrated in Figure 11. In particular, for the crop method, we find the smallest area that retains all the activated (>0) values in the heatmap and crop it. Table 3 demonstrates that IVM maintains robustness across all simple post-processing methods, where overlay+crop enjoys the most performance enhancement and thus is used as our default mask deployment method.

# 5 Conclusion

We introduce Instruction-guided Visual Masking (IVM), a generic and powerful visual grounding method that enhances broad multimodal instruction following tasks in a plug-and-play way. By masking out all instruction-irrelevant image regions, IVM effectively injects superior visual grounding ability to downstream LMMs non-intrusively, significantly boosting both commercial and open-sourced LMMs and achieving state-of-the-art results across numerous challenging multimodal benchmarks. Real robot experiments further demonstrate the versatility of IVM, showcasing the potential to deploy IVM to embodied robotic tasks where failures caused by distractions are long-standing challenges. For further improvement, one promising direction is to finetune LMMs using IVM-generated heatmap as an additional input channel to reduce suboptimal heuristics caused by mask deployment methods. Due to resource limitation, we leave this for future work. We open source the IVM checkpoint and the IVM-Mix-1M dataset to help the community further explore relevant directions[*]. More discussion on limitations and future directions can be found in Appendix A.

# 6 Acknowledgements

The paper is supported by funding from Wuxi Research Institute of Applied Technologies, Tsinghua University under Grant 20242001120. The authors would like to thank the anonymous reviewers for their feedback on the manuscripts.

---

[*]https://github.com/2toinf/IVM

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

# A  Limitation and Future Work

Here, we discuss our limitations, potential solutions and interesting future works.

1. **Computational Overhead**. Note that IVM introduces additional parameters and computational overhead to directly enhance visual grounding ability of LMMs, which in turn indirectly improve the VQA performances. However, more VQA performance gains can be obtained if the same amount of additional parameters are end-to-end trained directly on VQA data (LLaVA-13B v.s IVM-Enhanced LLaVa-7B in Table 2).

    *Solution and future work:* Nevertheless, this is quite reasonable because IVM primarily focuses on improving the visual grounding ability, but accurate VQA also requires other abilities which can be learned through end-to-end training. End-to-end training, however, requires tremendous VQA data to implicitly and slowly improve the visual grounding ability, which is quite data-intensive. Both Table 1 and Figure 1 can show that even trained on billions of data, GPT4-V still performs subpar on tasks that require strong visual grounding ability. IVM, instead, can significantly boost the visual grounding ability of GPT4-V using just 7B parameters and less computations. One promising and interesting future direction is to include some auxiliary tasks to directly absorb the strong visual grounding ability in the IVM-Mix-1M dataset through end-to-end training like [59].

2. **Data Quality**. Due to task complexity, the machine-annotated data in IVM-Mix-1M inevitably includes wrong labels that mistakenly exclude instruction-sensitive image regions or suboptimal labels that not fully mask out all instruction-irrelevant areas. These inaccuracies may lead to suboptimal IVM model. We propose a DWSL framework to tackle this. However, the DWSL framework relies on a learned discriminator and a human-designed $f(x)$ function, which may not exclude all inaccuracies.

    *Solution and future work:* We have clearly demonstrated in Figure 10 that with a simple $f(x)$ and a lightweight discriminator, DWSL consistently outperforms the naive Supervised Learning (SL), doing pretty well on prioritizing good samples and meanwhile identifying inaccurate labels. To further enhance this, one can use other advanced techniques such as Reinforcement Learning from Human Feedback (RLHF) [46, 5, 26] to provide more fine-grained judgement on annotation qualities, or resort a theoretical-soundness $f(x)$ [60] to achieve better results. In addition, one can also use our pretrained IVM model to directly generate high-quality heatmaps to enhance the machine annotations.

3. **Mask Deployment Methods**. In this paper, we directly use the simple post-processing method to apply the IVM generated heatmaps on images, which then are fed into LMMs to perform downstream tasks. However, these post-processing methods introduce some heuristics, which may be suboptimal for downstream LMMs. In addition, LMMs may not see many masked images during pretraining, thus some distributional shift may occur.

    *Solution and future work:* Although these limitations exist, IVM still obtain consistent improvements using diverse mask deployment methods, as shown in Table 3, which showcases the great versatility of IVM to inject visual grounding abilities. To further improve this, one strategy is employ some task-specialized visual search method [58], but will bring many computational load during inference and limit the versatility on embodied agents. Another promising direction is directly using the IVM generated heatmaps as an additional input channel to finetune the LMMs like [52], which can fully eliminate the heuristics of post-process methods, may bring larger performance gains. Due to resources limits, we leave this for a future work.

4. **Fine-grained Heatmaps**. Note that the IVM generated heatmaps cannot provide exact semantic object segmentation with clear contours like reasoning segmentation [31] offers.

    *Discussions:* We want to clarify that this is an advantage of the IVM model rather than a limitation. This is because of the ambiguous nature of the visual masking task. For this task, the ground truth heatmaps are mostly less semantic-meaningful for annotations as discussed in 3.1. So, we ensemble the annotation proposals from different visual grounding methods for data annotation, which will make the trained IVM model robust to include instruction-relevant image areas, rather than being aggressive to exclude some instruction-sensitive pixels like reasoning segmentation [31] does illustrated in Figure 2.

Overall, although some limitation exist, we have thoroughly discussed potential solutions to these limitations. Moreover, in this paper, we have demonstrated the superior effectiveness and versatility of IVM to directly inject strong visual grounding ability to downstream LMMs or embodied agents, representing a pioneer effort to extend traditional visual grounding methods towards a more complex and generic setting that covers diverse multimodal instruction following tasks.

## B    Broader Impact

This paper aims to advance the field of artificial intelligence, where no significant negative social impact is observed in this paper. The IVM-Mix-1M may contain some potential privacy issues and biases. However, in this paper, nearly all data are collected from open-sourced data, which have been well peer-reviewed, thus resolved this ethical concern.

## C    Training and Evaluation Details

### C.1    Architecture Details

In this section, we primarily focus on the architectural design of the lightweight generator and discriminator, as both the Language Model Multitask (LMM) and the vision backbone are derived from the powerful foundation models (LLaVA & SAM). Both the generator and discriminator utilize the same transformer-based decoder block, as depicted in Figure 12. We employ two such blocks for both the generator and discriminator. Specifically, the generator produces dense predictions by upscaling the output features of the decoder block through a straightforward upsampling operation. In contrast, the discriminator first employs a two-layer convolutional downsampling network to encode segmentation labels. This network, in conjunction with the decoder block and a simple MLP (multi-layer perceptron) head, outputs the weights.

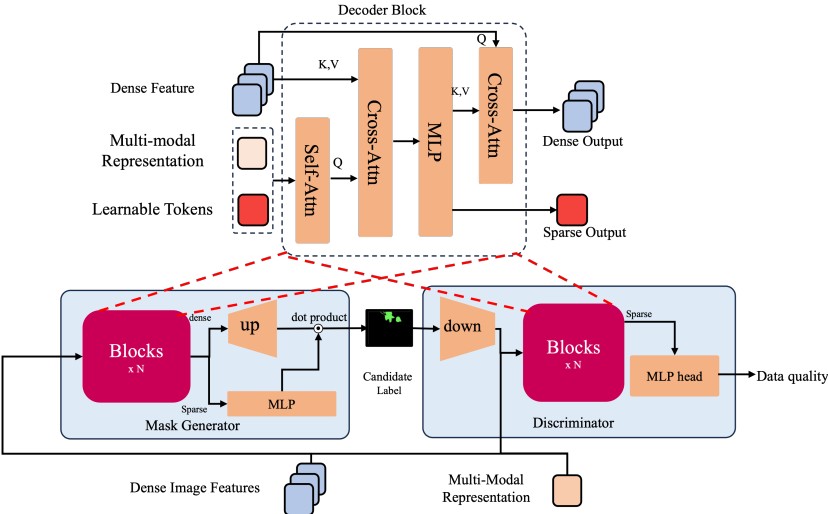

Figure 12: Generator/Discriminator Architecture Details

### C.2    Training Details

We adopt 8 NVIDIA 80G A100 GPUs and take 4 days to train our IVM model. The training scripts are based on deepspeed [4] engine and the training hyperparameters can be found in Table 4.

### C.3    Multimodal Benchmarks Evaluations

We evaluate our IVM on diverse multimodal benchmarks, including general VQA (VQAv2 [18], GQA [27], MME [16]), first-person perspective QA (EgoThink [10]), scientific QA (SQA [44]),

Table 4: Hyper-parameters for pretraining.

| config | value |
|---|---|
| training iteration | 200K |
| optimizer | AdamW [43] |
| learning rate | $1 \times 10^{-5}$ |
| batch size | 32 |
| weight decay | 0 |
| optimizer momentum | $\beta_1, \beta_2$=0.9, 0.95 |
| data augmentation | *RandomCropResize* |

hallucination adversarial QA (POPE [34]) and V* [58], a recently proposed challenging benchmark with high-resolution and complex visual input.

Our evaluation employs a two-stage inference pipeline: the image is firstly simplified by IVM-generated heatmap and mask deployment methods; Subsequently, the simplified image is fed into downstream LMMs(GPT4-V [1], LLaVA [38]) without finetuning. We adhere to the official procedures of each benchmark to evaluate the output of LMMs and report the results.

## C.4 Real Robot Evaluations

**Task descriptions**. The real robot experiments evaluate several `pick and place` manipulation tasks that require strong visual grounding abilities. Specifically, we evaluate on 4 tasks as shown in Table 5, following the task definitions in DecisionNCE [35]. For each task, we collect around 100 demonstrations using the demonstration collection system in BridgedataV2 [56]. We take both a side camera view and a wrist camera view as the vision inputs, as shown in Figure 13. For each demonstration, the environmental steps are around 50 steps. During data collection, the object and robot locations are randomly initialized, and the scene also has lots of randomly located distractors with varied shape and color.

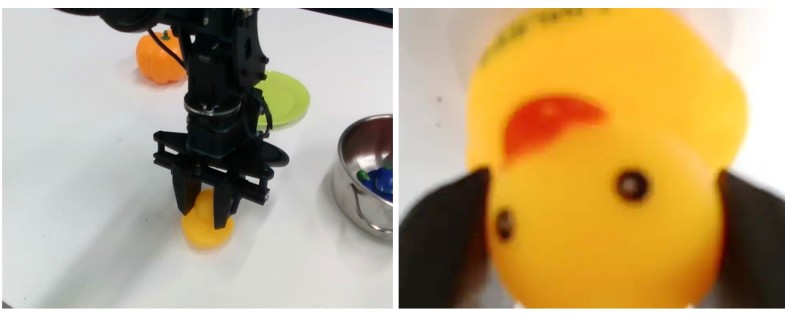

(a) WidowX RealRobot Side Camera View      (b) WidowX RealRobot Wrist Camera View

Figure 13: Visual input view for LCBC policy.

Table 5: Real Robot Tasks

| Environment ID | Language Instruction |
|---|---|
| Red cup on silver pan | Pick up the red cup and place it on the silver pan |
| Red cup on red plate | Pick up the red cup and place it on the red plate |
| Duck on green plate | Pick up the duck and place it on the green plate |
| Duck in pot | Pick up the duck and place it in the pot |

**Training details**. Here, we train Language-Conditioned Behavior Cloning (LCBC) policies using DDPM [24] loss since diffusion policies are good at fitting complex data distributions [67, 56, 2], especially human demonstrations [11]. For model architecture, the side and wrist images are augmented and then passed through a shared ResNet50 [22] image encoder and get an image

embedding for each camera view, following [56]. As the downstream data is quite limited, we load the ImageNet [13]-pretrained ResNet50 image encoder and further train it on the small robot data. Meanwhile, the language instruction is passed through a frozen T5 text-encoder [48], which is fused into the image encoder via Film conditioning layers [47]. Then, this language-conditioned image embedding is passed through a MLPs with residual connections similar to IDQL [21], which then outputs the predicted noise in DDPM [24]. To obtain smoothed policy rollouts, we adopt Action Chunking and Temporal Ensemble from ACT [66] with a chunking size 4 rather than 100 in [66] because the episode horizons in this paper are only around 50. The LCBC policies are trained either on the original side camera view (without IVM assistance) or on the IVM-masked side camera view (with IVM assistance) for 200K steps with a batch size of 64. The training can be completed on 2 NVIDIA RTX4090 GPU in 17h. All hyperparameters are summarized in Table 6.

Table 6: Real robot LCBC training details

| Backbones | |
| --- | --- |
| Visual encoder | Resnet50 [22] (ImageNet [13] pretrained) |
| text encoder | T5 [48] (frozen) |
| **DDPM hyperparameters** | |
| noise schedule | VP [51] |
| denoising time steps | 25 |
| **Other hyperparameters** | |
| Chunking size | 4 |
| Optimizer | AdamW [43] |
| Learning rate | 1e-4 |
| Lr schedule | cosine annealing |
| Warm up steps | 2000 |
| Batch size | 64 |
| Gradient Steps | 200K |
| Augmentation | Yes [56] |

**Evaluation details**. We first evaluate the trained LCBC policies without strong distractions, where no or only small distractors appear in the image. Then, we add lots of distracting objects with varied shapes and colors, and even introduce strong human disturbance to attack the LCBC policies. For each score reported in Figure 9, we evaluate 10 episodes and report the success rates.

## D   Mixture of Expert Annotation Pipeline

### D.1   Labeled Visual Grounding data

For labeled visual grounding data, We provide the following prompt to drive GPT-4 [1] to generate more complex instructions based on given language annotations.

[Image Description] %s

[System] You are an AI visual assistant, and you are seeing a single image. What you see are part of the image and are provided with a simple phrase. Please generate any instructions that can be executed based on the content of the picture described, including simple queries about the content of the picture, such as the object types, counting the objects, object actions, relative positions between objects, etc. Also consider more complex questions that require reasoning. For example, you can ask what time it is now for a clock and what can I use to clean the room for a broom. Ensure that the questions you ask can be clearly answered only based on what you see. Please generate as many five questions as possible and return them in a single line separated by ';' and avoid any other output.

## D.2 Unlabeled Visual Instruction Following Data

For unlabeled visual instruction following data, we first try to simplify complex instructions. Specifically, we employ GPT-4 to infer the necessary object for executing the given instructions based on these instructions and a simple image caption. If the dataset lacks captions, they can be generated using an existing caption model like BLIP-Caption [33]. Below, we outline the prompts specifically designed for GPT-4.

> [Image Caption] %s
>
> [Instruction] %s
>
> [System] You are an helpful AI assistant. I need to reply to the previous instruction based on an image, and I have a simple caption for the image. Please note that there may be objects in the image that I did not detect. Since you cannot view the image, please list any potential objects that might influence my responses, separated by semicolons, in a single line without any additional output. If you believe that the number of objects could be too extensive and might hinder my judgment, print 'None'.

With the simplified instruction, we can adopt existing visual grounding models to generate the candidate label. Specifically, we utilize four models: AlphaCLIP [52], LISA [31], OwVIT [20] and Grounding-SAM [49] and the inference pipelines are provided in the official implementation of these models.

# E More result

## E.1 Referring Expression Comprehension

As IVM is an extension of traditional visual grounding task, we also evaluate our IVM on RefCoCo, RefCoCo+ and RefCoCog [64]. We reported the accuracy (IOU-50%) on the validation split in Table 7. As a generalist model capable of handling complex instructions, our IVM achieves performance comparable to that of state-of-the-art (SOTA) specialist models.

Table 7: result in REC

| Methods | RefCoCo | RefCoCo+ | RefCoCog |
|---|---|---|---|
| *Specialist models* | | | |
| G-DINO-L [39] | 90.56 | 82.75 | 86.13 |
| *Generalist models* | | | |
| LLaVA-7B [41] | 76.29 | 66.76 | 70.4 |
| IVM(Ours) | **90.1** | **83.3** | **82.9** |

## E.2 Visualization Result

In this section, we provide more visualization result in VQA-type data as shown in Figure 14.

**Failure Case**. Although we observe numerous successful instances, our IVM still faces significant challenges, as illustrated in Figure 15. We summarize these challenges into three categories: missing target, misguided target, and insufficient reasoning.

(a) **Missing Target**: Challenges arise when target objects are relatively small and scattered around many separate image corners. In this case, accurately detecting all of the targeted objects is quite difficult. Even specialized open vocabulary detection models struggle with this task. For example, the cup on the right in the image is masked by the IVM mistakenly. However, we still observe that the IVM-generated heatmap for the right cup is partially activated, meaning that IVM have partially focus this regions. We believe by providing more training data, IVM can handle this better.

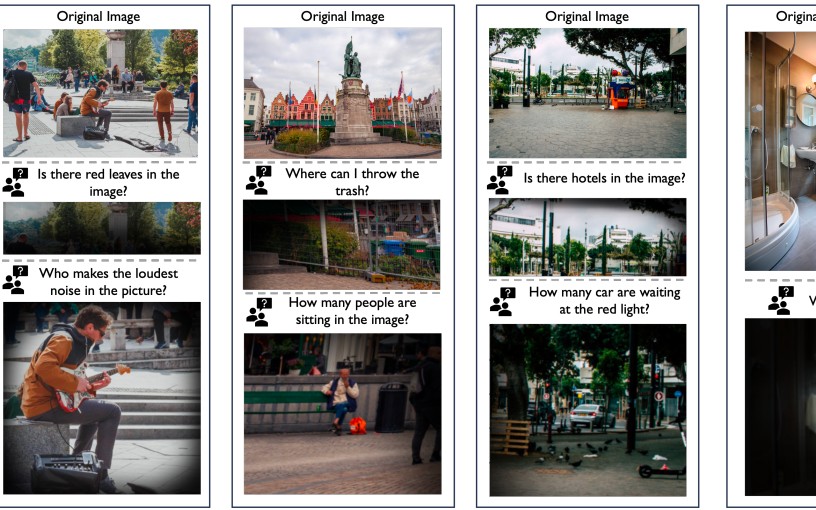

Figure 14: Visualization results of IVM generated masks.

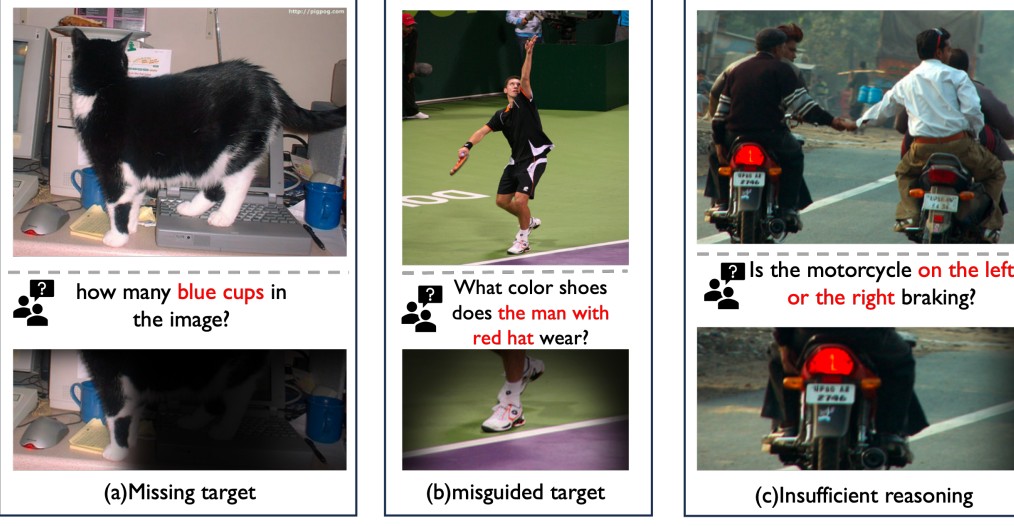

Figure 15: Some failure cases.

(b) **Misguided Target**: Accurately Localizing tiny target objects is a recognized challenge [58], especially when similar but more obvious objects are present. For instance, IVM incorrectly focuses on the more centrally located shoes of another man, instead of the shoes of the man wearing the **red hat** at the edge of the picture. However, this instruction is pretty challenging that at first glance, even a human might struggle to spot the man with the red hat in the left corner. We will leave challenge scenarios like this for future research.

(c) **Insufficient reasoning**: The objective of the IVM task is to assist LMMs in extracting visual features more effectively to better follow instructions. Thus, the demands on the model's reasoning capabilities extend far beyond mere object localization. Although IVM demonstrates strong performance, it sometimes overlooks additional image content necessary for accurately following instructions after correctly locating the target object. For instance, while IVM successfully identified the braking motorcycle, it failed to recognize that answering the question requires knowledge of the positions of both motorcycles simultaneously. We attribute this issue to biases in the training data. By incorporating more complex instructions and diversified labels, we anticipate that our model will achieve improved performance

### E.3 Robotics Result

Here, we provide more evaluation rollouts of the IVM-assisted LCBC agents under strong distractions. Figure 16 clearly demonstrates that even under strong distractions like the background are full of distracting objects with similar colors or shapes to the targeted objects, and strong human disturbances that adversarially attack the robots, the IVM-assisted LCBC agents can still complete the tasks pretty well, enjoying high-level of generalization and robustness thanks to the superior visual grounding ability injected by IVM. More videos can be found in the supplementary materials.

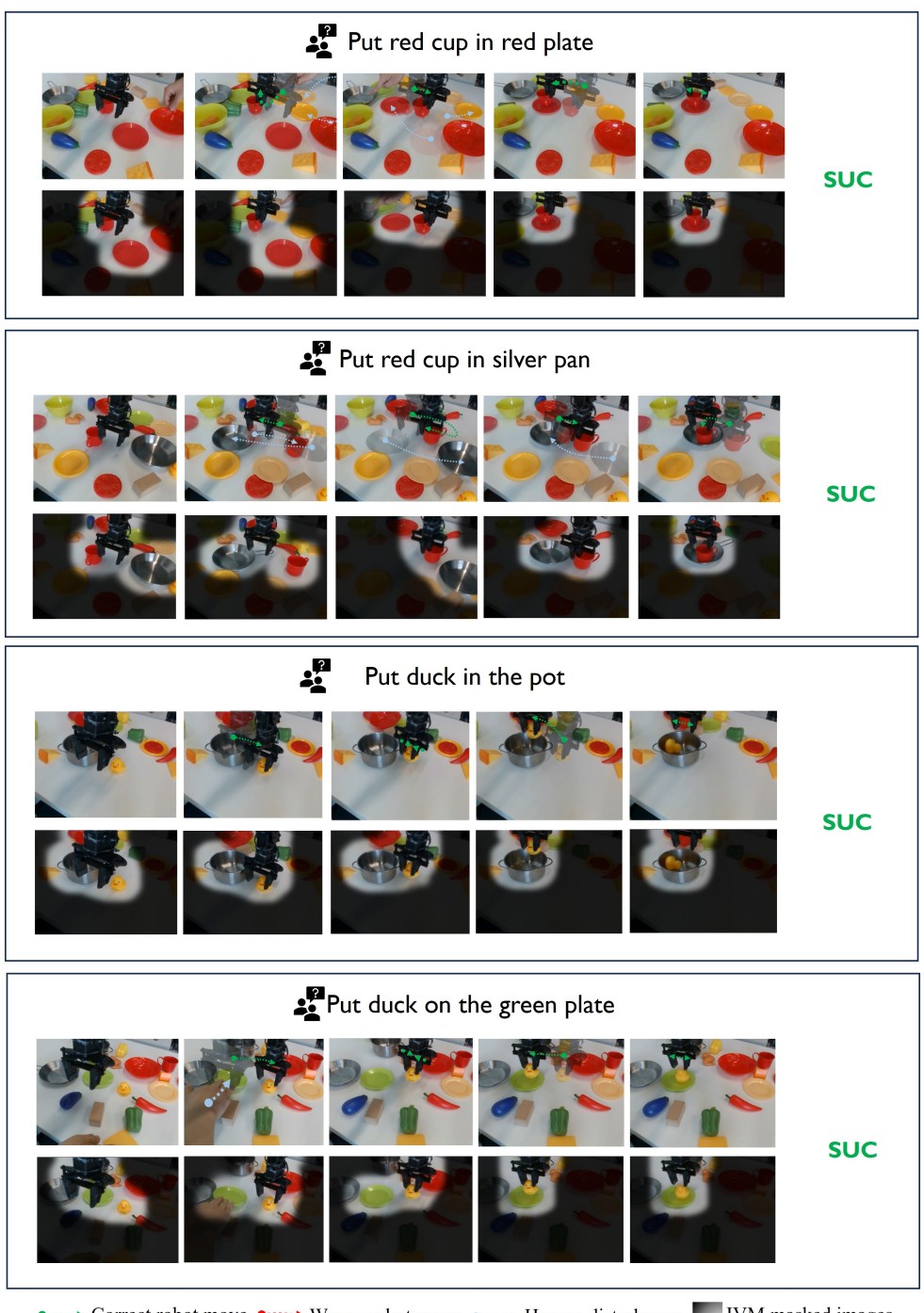

Figure 16: Real robot LCBC results with IVM assistance.

