# OpenReview forum: "Instruction-Guided Visual Masking"
_NeurIPS.cc/2024/Conference — NeurIPS 2024 poster_

### Official Review · Reviewer_zLHv · 2024-07-02

**Soundness:** 3
**Presentation:** 3
**Contribution:** 3
**Rating:** 6
**Confidence:** 4

**Summary:**

In this paper, the author introduce Instruction-guided Visual Masking (IVM), a generic and powerful visual grounding method that enhances broad multimodal instruction following tasks in a plug-and-play way. By masking out all instruction-irrelevant image regions, IVM effectively injects superior visual grounding ability to downstream LMMs non-intrusively, significantly boosting both commercial and open-sourced LMMs and achieving state-of-the-art results across numerous challenging multimodal benchmarks.

**Strengths:**

Overall, the motivation of this paper is commendable, as it addresses an interesting problem and provides rich illustrations that facilitate understanding. Furthermore, the work is supported by a substantial number of experiments to validate its effectiveness.

**Weaknesses:**

1. The routine of retraining models by preparing new datasets, as described in this paper, can be considered somewhat old-fashioned.
2. The technical insight of this paper is relatively weak, and no strong novelty technical methods are proposed.

**Questions:**

I have a few minor concerns that I would like the author to address.

1. The routine of retraining models by preparing new datasets, as described in this paper, can be considered somewhat old-fashioned. Recent literature has proposed similar grounding datasets, such as Ferret (GRIT) [1] and Kosmos-2 (GRIT) [2]. However, these two works are not cited. There should be some new discussion on motivation about the dataset in this paper.

2. Why does the instruction-related area between 90% and 100% in Figure 5 increase abnormally significantly? It is necessary to consider whether these data needs to be cleaned.

3. Formulas 1 and 2 in the paper are somewhat deliberately mystifying. I understand that the purpose is to learn an optimal model from the hand-labeled data and the auto-generated data. This process is somewhat similar to the pseudo-label curriculum learning proposed in CLIP-VG [3]. It is suggested to increase relevant discussion and citations.

4. I have some confusion regarding Figure 6 of the paper, which requires modification. Specifically, (1) the discriminator and generator should be two separate processes, while in Figure 6, the image, text, and attention are depicted as being fed to the model simultaneously; (2) It is unclear whether the LLM in the figure represents one model or two models since there are two LoRAs shown but only one LMM is illustrated.

5. A framework diagram of the inference model used in the downstream experiment should be drawn according to Figure 6, so as to illustrate how the IVM assists the model in inference.

6. Since the paper claims that IVM is a plug-and-play method for assisting LLM, so I am curious about how this paper performs on the RefCOCO/+/g dataset after incorporating IVM. After all, the motivation behind this paper stems from the visual grounding task, and there has been a number of research on grounding multimodal large language models, such as Ferret [1], Kosmos-2 [2], and LION [4].

7. Other writing issues, such as some vocabulary in the paper should be consistent. For example,
(a) line 135 RefCoCo should be changed to RefCOCO to maintain uniformity;
(b) Lora is used in Figure 5, while LORA is used in the main text, and it is recommended to use LoRA uniformly in order to be consistent with the original article;
(c) LMM and LLM are used in this paper, however LLM has not the full vocabulary,  and it is suggested maybe better to use LLM and MLLM uniformly. These rudimentary errors should not appear in NeurIPS submission papers.

On the whole, this paper is a relatively solid work, the overall presentation is good, literature review is relatively sufficient, so I currently give a positive scores. Hope the author can address my concerns, and I will decide whether to raise down or raise up my rating according to the author's rebuttal reply.

--

[1] You, Haoxuan, et al. "Ferret: Refer and Ground Anything Anywhere at Any Granularity." The Twelfth International Conference on Learning Representations. 2024

[2] Peng, Zhiliang, et al. "Kosmos-2: Grounding multimodal large language models to the world." arXiv preprint arXiv:2306.14824 (2023).

[3] Xiao, Linhui, et al. "CLIP-VG: Self-paced Curriculum Adapting of CLIP for Visual Grounding." IEEE Transactions on Multimedia (2023).

[4] Chen, Gongwei, et al. "Lion: Empowering multimodal large language model with dual-level visual knowledge." Proceedings of the IEEE/CVF Conference on Computer Vision and Pattern Recognition. 2024.

**Limitations:**

see weaknesses.

---

> ### Author Rebuttal · Authors · 2024-08-02
>
> We sincerely appreciate the reviewer for the positive feedback and the constructive comment on our work! Regarding the concerns from the reviewer zLHv, we provide the responses as follows:
>
> >**W1: The routine of retraining models by preparing new datasets, as described in this paper, can be considered somewhat old-fashioned.**
> - No, our work introduces a novel and meaningful task setting, instruction-guided visual masking, and finds that all existing methods and data are insufficient for training models effectively in this new context. Thus, constructing suitable data is the most straightforward, effective, and crucial approach to solving this problem, not outdated.
> - Moreover, we also introduced a DWSL training objective, which is essential for the IVM model's success. Without DWSL, Figure 9 demonstrates that the naive supervised learning fails miserably when training on the IVM-Mix-1M data.
>
> >**W2: The technical insight of this paper is relatively weak, and no strong novelty technical methods are proposed.**
> - No, as noted above, the instruction-guided visual masking task offers a new perspective and approach to improving multimodal instruction following abilities.
> - Also, the DWSL approach provides an effective solution for leveraging mixed-quality data to train a good model.
>
> >**Q1: The routine of retraining models by preparing new datasets can be considered somewhat old-fashioned. Recent literature has proposed similar grounding datasets, such as Ferret (GRIT)[1] and Kosmos-2 (GRIT)[2]. However, these two works are not cited. There should be some new discussion on motivation about the dataset in this paper.**
> - Thanks for bringing these insightful works to our attention! We will include all these works in future versions of our work. Here we provide a brief discussion of their core differences from our work:
> - The GRIT dataset is fundamentally different from our IVM-Mix-1K dataset in several key aspects:
>     - `Label Granularity`: IVM aims to mask out instruction-irrelevant visual contents, necessitating pixel-level predictions instead of the simple instance-level detection in GRIT. This significantly raises the data demand.
>     - `Data Source and Construction Method`: Most data in the GRIT dataset originate from sources already equipped with grounding labels, which are then transformed into instruction-following data using LLMs akin to the first part of our proposed pipeline in Figure 4 (1). In contrast, lots of IVM-1M-Mix data derive from unlabeled data with a broader range of instructions, such as robot learning data.
>
> >**Q2: Why does the instruction-related area between 90% and 100% in Figure 5 increase abnormally significantly? It is necessary to consider whether these data needs to be cleaned.**
> - No, this data is essential and should not be cleaned, since many instructions require a comprehensive understanding of all the visual input without specific visual grounding. For example, all caption-style instructions like "Describe the image," "Where am I," or "Write a poem according to the image" should focus on the entire image rather than specific image areas (see Figure 7 for details). So, the model must learn to respond accurately to these complex instructions without losing contextual information.
>
> >**Q3: Formulas 1 and 2 in the paper are somewhat deliberately mystifying... This process is somewhat similar to the pseudo-label curriculum learning proposed in CLIP-VG [3]. It is suggested to increase relevant discussion and citations.**
> - Thanks for bringing this insightful work to our attention! We are very happy to discuss relevant works in future revisions! Here, we provide a brief discussion:
>     - Pseudo-label curriculum learning in CLIP-VG differs from our DWSL objective. DWSL can be regarded as a weighted regression objective, where all labels are pre-determined and fixed. In contrast, CLIP-VG is more like semi-supervised learning where the labels are gradually updated and thus may suffer from compounding errors in the iterative cycles.
>
> >**Q4: I have some confusion regarding Figure 6 of the paper, which requires modification...**
> - We apologize for any confusion caused by the original figure and appreciate this constructive feedback! We have updated Figure 6 (modified) in the PDF attached with **General Response**.
>
> >**Q5: A framework diagram of the inference model used in the downstream experiment should be drawn according to Figure 6, so as to illustrate how the IVM assists the model in inference.**
> - Thanks for this helpful suggestion! We have added the inference pipeline in Figure a in the PDF attached with **General Response**. Very happy to hear any further comments!
>
> >**Q6:I am curious about how this paper performs on the RefCOCO/+/g dataset after incorporating IVM...**
>
> - Yes, indeed, the comparison with VG methods on the RefCOCO-series benchmarks can be found in Table 7 in Appendix E.1, where IVM demonstrates comparable results to SOTA specialist VG models and outperforms other generalist models.
> - Here, we also compare IVM with the mentioned KOSMOS-2, LION, Ferret and Shikra[1]. Results show that, as a plug-and-play tool, IVM achieves strong results compared to other specially designed or trained baselines. Note that we report the result on the validation split of each dataset here and the '\*' denotes zero-shot performance.
>
> |Methods|RefCOCO↑|RefCOCO+↑|RefCOCOg↑|
> |---|---|---|---|
> |KOSMOS-2*|52.32|45.48| 60.57|
> |LION-4B|89.73|83.60|85.69|
> |Shikra-7B[1]|87.01|81.60|82.27|
> |Ferret-7B|87.49|80.78|83.93|
> |IVM-7B(ours) |90.1|83.3|82.9|
> - Thanks for mentioning these great works again! We will include these comparisons in future versions of our work.
>
> >**Q7:Other writing issues...**
> - Thanks for the efforts and these helpful comments! We apologize for the rudimentary errors found in the article. We will conduct a thorough review and correct them in future revisions!
>
> [1]Shikra: Unleashing multimodal llm's referential dialogue magic, 2023

---

> > ### Comment · Reviewer_zLHv · 2024-08-12
> > **The author's response is polite, and I appreciate the author's efforts during this rebuttal process.**
> >
> > I noticed that the author did not fully respond to some of my questions, such as explaining equations 1 and 2 in Q3, etc. Additionally, the authors' results in Q6 appear to be insufficient compared to several baselines. Nevertheless, Considering the author's response is polite, and I appreciate the author's efforts during this rebuttal process. Therefore, I would like to raise my rating by 1 point to `weak accept`. I expect the authors to make corresponding revisions regarding the aforementioned issues in the final version.

---

> ### Author Response · Authors · 2024-08-12
> **Thanks for increasing the score!**
>
> We really thank the reviewer for increasing the score, and sorry for the confusion in the rebuttal phase! Regarding the remaining concerns, we provide the following detailed responses.
>
> >**Explanation about Eq. (1) and Eq.(2)**
>
> - Due to the space limits in the rebuttal phase (6000 characters), we did not include very detailed discussions on Eq. (1) and Eq. (2) in the rebuttal. The introduction of Eq.(1) and Eq. (2) is inspired by recent offline IL work [2] that tries to learn a good model jointly from a high-quality near-expert dataset (human data) and a mixed-quality suboptimal dataset (machine-generated data), which typically consists of two stages:
>
>    - `Discriminator Training`. Eq. (1) tries to train a discriminator $d$ to distinguish between human data $\mathcal{D}_E$ and the machine-generated data $\mathcal{D}_D$. By doing so, the output of the discriminator $d$ becomes a confidence value for data quality ($d$ will output near 1 or near 0 if the data can be clearly identified as human or machine data; $d$ will output around 0.5 if the annotation quality is hard to judge), see Figure 9 (b) for discriminator output statistics.
>
>    - `Discriminator-weighted Supervised Learning`. After training the discriminator, the $d$ value can be an adaptive weight function to reprioritize the training annotations in Eq. (2), where high-quality annotations will be more fitted than low-quality ones. In this sense, the side-effect of bad annotations in the mixed-quality machine data can be largely filtered out, see Figure 9 (a) for detailed experimental evidence.
>
> Here, all annotations are pre-collected rather than gradually updated during training as CLIP-VG does. So, we're more insensitive to the compounding errors than CLIP-VG.
>
> >**Additional discussions on Q6**
>
> - We are really sorry for the confusion here. Due to time limits, we tested IVM only on the validation split rather than all refcoco splits.  Now, we're trying to implement IVM-13B by scaling IVM on more human and machine data. After that, we will conduct more thorough experiments in the future.
>
> >**I expect the authors to make corresponding revisions regarding the aforementioned issues in the final version.**
> - Sure! We will consider all these helpful suggestions when revising our paper!
>
> Thanks to the reviewer again for the efforts and engagement in the discussion phase, and open to any further comments!
>
> [2] Discriminator-weighted offline imitation learning from suboptimal demonstrations. ICML 2022

---

> > ### Comment · Reviewer_zLHv · 2024-08-12
> > **Thanks for the authors' responses. That's all.**
> >
> > Thanks for the authors' responses. That's all.

---

### Official Review · Reviewer_EbuK · 2024-07-12

**Soundness:** 4
**Presentation:** 2
**Contribution:** 3
**Rating:** 6
**Confidence:** 4

**Summary:**

This paper introduces the Instruction-guided Visual Masking (IVM), a versatile visual grounding model designed to improve alignment between textual instructions and specific image regions. It outlines the development of a visual masking data generation pipeline and a new learning technique, Discriminator Weighted Supervised Learning (DWSL), which prioritizes high-quality data samples to enhance performance on multimodal tasks.

**Strengths:**

1. This paper introduces instruction-guided visual masking (IVM). The IVM-enhanced multimodal models can focus on task-relevant image regions to better align with complex instructions. This implies that the model can become more sensitive to instructions.
2. Figures 2 and 3 are helpful to understand the method.
3. This paper has collected a richer and more complex visual grounding dataset.

**Weaknesses:**

1. The IVM model architecture shown in Figure 6 is not conducive to understanding the approach.
2. It is recommended to compare more methods on a multimodal benchmark.

**Questions:**

1. The paper mentions that the IVM-based model does not show significant gains on the GQA, SQA, and VQAv2 benchmarks. It suggests that these benchmarks may not depend heavily on grounding abilities, which raises some doubts for me. Intuitively, the results for simple visual inputs related to instructions should be better.
2. Additionally, I am concerned that the improved performance of the proposed method over comparative models might largely be attributed to the use of a larger Visual Grounding dataset. I am interested in seeing how other models would fare if they were trained or fine-tuned using the same dataset.

**Limitations:**

Limitations and broader impact have been discussed.

---

> ### Author Rebuttal · Authors · 2024-08-02
>
> We sincerely appreciate the reviewer for the positive feedback and the constructive comment on our work! Regarding the concerns from the reviewer EbuK, we provide the responses as follows:
>
> >**W1: The IVM model architecture shown in Figure 6 is not conducive to understanding the approach.**
> - Sorry for any confusion caused by the original figure! We have updated Figure 6 (modified) in the PDF attached with **General Response**. Very happy to hear any further comments!
>
> >**W2: It is recommended to compare more methods on a multimodal benchmark**
> - Thanks for this helpful suggestion! Given that we use the llava-1.5 model as our base model, we initially reported only the six most classic baselines from the same period in Table 2 to ensure a fair comparison. We will include additional baselines to more clearly demonstrate IVM's performance in future versions.
>
> >**Q1: The paper mentions that the IVM-based model does not show significant gains on the GQA, SQA, and VQAv2 benchmarks. It suggests that these benchmarks may not depend heavily on grounding abilities, which raises some doubts for me. Intuitively, the results for simple visual inputs related to instructions should be better.**
> - No, in scenarios with simple visual inputs, the bottleneck is no more visual grounding but other abilities like reasoning, etc, since almost all visual contents are instruction-relevant (see Figure 7 in our paper for details ).  Thus, the IVM-based model does not show significant gains on the simple GQA, SQA, and VQAv2 benchmarks that do not require strong visual grounding abilities (we provide some examples in Figure b in the PDF attached with General Response), but shows superior advancements on the challenging V* benchmark.
>
>
> >**Q2: Additionally, I am concerned that the improved performance of the proposed method over comparative models might largely be attributed to the use of a larger Visual Grounding dataset. I am interested in seeing how other models would fare if they were trained or fine-tuned using the same dataset.**
>
> - No, the DWSL training objective is also very crucial for the improvements. Figure 9 (a) demonstrates that DWSL is the only method that can effectively leverage both the high-quality but small human data and the large but mixed-quality machine data to achieve superior results. The simple supervised training baseline, however, fails miserably in training solely on machine or human data.
> - Also, note that the supervised learning baseline degenerates to the previous method LISA [1] as we follow the architectural framework established by LISA. So simply finetuning baseline models using the same dataset is not enough to get good results. DWSL, however, can enjoy considerable improvements.
>
> [1] LISA: Reasoning Segmentation via Large Language Model, CVPR 2024

---

> ### Comment · Reviewer_EbuK · 2024-08-12
>
> I feel that my concerns were not fully addressed. The explanations provided seem more intuitive and lack solid theoretical or experimental support, such as Q1 and Q2. Given this, I decided to lower my rating to weak accept.

---

> > ### Author Response · Authors · 2024-08-12
> > **Thanks for the further comments**
> >
> > We thank the reviewer for the efforts and engagement in the discussion phase. Regarding the remaining concerns, we provide additional responses as follows:
> >
> > >**Additional responses to Q1**
> > - For Q1, we provide the analysis of the relationships about data quantities w.r.t different `instruction-relevant visual ratios (IVR)` （the ratios of preserved pixels in the original images） generated by IVM on different multimodal benchmarks including the simple GQA, SQA, VQAv2 (due to time limits, we evaluated on 1/10 subset of the VQAv2 dataset, if the reviewer would like to see results on full datasets, we will try our best to finish the experiments by the author rebuttal deadline) and the more complex V* and EgoThink benchmarks.
> >
> > |Benchmark|<20% (IVR) |20%-40% (IVR)|40%-60% (IVR)|60%-80% (IVR)|>80% (IVR)|
> > |---|---|---|---|---|---|
> > |V*|100%|0%|0%|0%|0%|0%|
> > |Egothink| 67% | 14% | 4% | 3% | 12%|
> > |VQAv2 (40K samples)| 13% | 16% | 12% | 10% | 49%|
> > |GQA| 17% | 14% | 8% | 4% | 57% |
> > |SQA| 0% | 5%| 11% | 7% | 77% |
> >
> > - These statistics demonstrate that most visual contents in GQA, SQA and VQAv2 benchmarks are instruction-relevant (>80% IVR). Existing base MLLMs can easily focus on the correct image areas to follow the instructions, rather than be distracted by the minor visual distractors (for example, see Figure b in the PDF in **General Response**).
> > - On the contrary, only a small ratio of visual contents in the challenging V* and Egothink benchmark are instruction-relevant (<20% IVR). In this case, MLLMs are more likely to be distracted by a lot of irrelevant visual content if the MLLMs' visual grounding ability is not that strong. With the IVM assistance, however, the performance can be greatly enhanced via the explicit surgical visual grounding.
> >
> > >**Additional responses to Q2**
> > - For Q2, `LISA is the only comparable baseline in our setting`. Specifically, most MLLMs cannot be directly finetuned or trained in our setting as they are not primarily designed to predict dense image heatmaps given complex instructions that require strong reasoning ability. Among all these MLLMs, LISA is the most relevant work that can be fairly compared, as LISA is specifically tailored for reasoning segmentation, which shares some similarity with our setups.
> >
> > Indeed, we have empirically compared the baseline method LISA (please check the Human+machine label (SL) baseline in Figure 9 (a) in our paper for details). Here, we summarize the results more directly (detailed comparisons of diverse data quantities and data components can be found in Figure 9 (a)).
> >
> > |Training object & data|IVM on IVM-Mix-1M (Ours)|LISA on IVM-Mix-1M|
> > |---|---|---|
> > |Improvements on V* benchmark|**+26.2**|+16.2|
> >
> > `If the reviewer has any further detailed concerns or requires any other experimental supports, please do not hesitate to point them out. We will be more than happy to address them and improve the quality of our paper.`

---

> ### Author Response · Authors · 2024-08-13
>
> Hi! The discussion is approaching to close. So, hope our additional responses successfully address your remaining concerns. If not, please do not hesitate to point them out. We would really appreciate any further comments that can improve the quality of our paper!

---

### Official Review · Reviewer_bwCu · 2024-07-13

**Soundness:** 3
**Presentation:** 3
**Contribution:** 4
**Rating:** 8
**Confidence:** 5

**Summary:**

This paper presents IVM (Instruction-guided Visual Masking). The key idea is that we could mask out the instruction-irrelevant regions in the given image. The trained model is tasked to mask out the irrelevant regions, enforcing the multimodal model to focus on the task-related visuals. Such grounding-centric approach is effective in enhancing different multimodal models.

The paper also details their solution to create a large number of reliable pixel-level labels. The paper presents a MoE pipeline with various visual grounding models to collect reliable labels.

The paper further introduces DWSL for IVM training. DWSL helps IVM training to focus more on higher quality training data.

**Strengths:**

- The proposed IVM is sound and effective. I found the problem formulation of generating heatmap interesting. The experiments clearly demonstrate the effectiveness of the proposed method.
- The MoE pipeline is well-developed and can be applied to both labeled and unlabeled data.
- The IVM architecture with discriminator training can effectively reduce the impact of low-quality data.
- The paper is well-written and solid.

**Weaknesses:**

I don't find significant problems in the paper. One possible improvement for this paper is that it would be interesting to provide some typical failure cases of the proposed method.

**Questions:**

see weakness

---

> ### Author Rebuttal · Authors · 2024-08-02
>
> We sincerely appreciate the reviewer for the positive feedback and the constructive comment on our work! Regarding the concerns from the reviewer bwCu, we provide the responses as follows:
>
> >**W1: I don't find significant problems in the paper. One possible improvement for this paper is that it would be interesting to provide some typical failure cases of the proposed method.**
> - Thanks for this suggestion! Indeed, several failure cases along with detailed discussions can be found in Figure 14 in Appendix E.2. These examples are intended to deepen the understanding of the IVM model and inspire future improvements.

---

> > ### Comment · Reviewer_bwCu · 2024-08-10
> > **comment**
> >
> > Thank you. I have no further comment at this point.

---

> > > ### Author Response · Authors · 2024-08-12
> > > **Thanks for the responses!**
> > >
> > > Thanks to the reviewer again for the efforts and engagements, and open to any further comments!

---

> ### Comment · Reviewer_zLHv · 2024-08-12
> **I am not convinced by reviewer bwCu's comments and rating, as it seems that the reviewer bwCu was aware of the author's identity and deliberately gave high marks.**
>
> After reading the comments of the other reviewers, I am not convinced by reviewer bwCu's comments and rating. To be specific,
>
> - (a) The reviewer bwCu claims that his confidence in this paper is "absolutely certain about your assessment. You are very familiar with the related work." However, the reviewer bwCu did not give some valuable and in-depth comments to this paper. I think, if the reviewer bwCu is familiar with this field, he should be able to find the defects of this paper and make some valuable comments.
> - (b) Reviewer bwCu gave nothing but praise for this paper;
> - (c) Judging from the comments of the other three reviewers, there are still more or less problems in this paper. However, reviewer bwCu directly gave the extreme rating of "Strong Accept".
>
> To sum up, it seems that the reviewer bwCu was aware of the author's identity and deliberately gave high marks.

---

> > ### Comment · Reviewer_EbuK · 2024-08-12
> >
> > I agree with reviewer zLHv's perspective on the concerns regarding reviewer bwCu's comments and rating.
> > Compared to the comments from other reviewers, reviewer bwCu's comments seem overly positive and lack the professional suggestions expected.
> > This raises my curiosity about the reason behind the reviewer bwCu's strong accept rating.

---

### Official Review · Reviewer_5SdR · 2024-07-14

**Soundness:** 3
**Presentation:** 3
**Contribution:** 2
**Rating:** 6
**Confidence:** 3

**Summary:**

For the purpose of precise instruction following performance in LLM, this paper proposes a versatile grounding model that is compatible with diverse multi-modal models. Leveraging the LLM, a visual masking data generation pipeline is built and 1 million image-instruction pairs are constructed. On top of it, an Instruction-Guided Visual Masking (IVM) model is trained to focus on task-relevant regions that align with complex instructions.

**Strengths:**

1. An IVM model is proposed to enhance multimodal instruction following via nuanced surgical visual grounding. Overall, this model is simple yet effective. Such a model can be seamlessly incorporated into a multimodal model to boost the performance of downstream tasks.
2. A dataset creation pipeline with a mixture of experts is carefully devised and along with it an 1-M dataset is built.
3. To effectively utilize the dataset, a discriminative weighted supervised learning training strategy is devised to select the high-quality dataset pairs.
4. Extensive experiments have been conducted to validate the effectiveness of the proposed approaches on various tasks, e.g. visual understanding, reasoning segmentation model and real robot control.

**Weaknesses:**

1. This work proposes a visual grounding model. However, it seems that the comparison with state-of-art visual grounding methods (VG) is missing. I understand this task involves slight differences with reasoning segmentation (RS) tasks. But will the VG task evaluation be more direct?
2. This approach relies heavily on the human manually labeled data, without which the performance will significantly drop. Therefore, it is more like a dataset creation work. Finetuning on this dataset, other compared works might also achieve similar or even better performance.
3. The derived framework is a little bit heavy, with LLM and heavy visual rendering head involved . There might be a more efficient network framework to achieve similar performance.

**Questions:**

Refer to the questions in weakness section.

Other questions related to technical details.
1. How does this approach adapt to images with distinct image resolution? Is there any specifically designed relative position embedding?
2. Is there any plan to extend this work to more complicated robot action? The pick and put are very simple action. The real world application will involve more convoluted operations such as visual navigation or sequential manipulation.

**Limitations:**

Authors adequately addressed the limitations.

---

> ### Author Rebuttal · Authors · 2024-08-02
>
> We sincerely appreciate the reviewer for the positive feedback and constructive comments on our work! Regarding the concerns from the reviewer 5SdR, we provide the responses as follows:
>
> > **W1: It seems that the comparison with visual grounding methods (VG) is missing. Will the VG task evaluation be more direct?**
>
> - Indeed, the comparison with VG methods on the RefCOCO-series benchmarks can be found in Table 7 in Appendix E.1, where IVM demonstrates competitive results to SOTA specialist VG models and outperforms other generalist models.
> - Here, we provide more comparisons to other baselines that are specifically designed to ground multimodal large language models. Results show that, as a plug-and-play tool, IVM achieves strong results compared to other specially designed or trained baselines. Note that we report the result on the validation split of each dataset here and the '\*' denotes zero-shot performance.
>
> | Methods | RefCOCO↑ | RefCOCO+↑ | RefCOCOg↑ |
> | -------- | -------- | -------- | -------- |
> | KOSMOS-2*[1]| 52.32 |  45.48| 60.57 |
> | LION-4B[2]|89.73|83.60|85.69|
> | Shikra-7B[3]| 87.01 | 81.60 | 82.27 |
> | Ferret-7B[4]| 87.49 | 80.78 | 83.93 |
> | IVM-7B(ours) | 90.1 |83.3 |82.9   |
>
> > **W2: This approach relies heavily on the human manually labeled data... Therefore, it is more like a dataset creation work. Finetuning on this dataset, other compared works might also achieve similar or even better performance.**
>
> - No, our work extends beyond mere dataset creation. Firstly, we introduce a novel and groundbreaking setting, instruction-guided visual masking, where all previous methods failed under this setting. To address this issue, we further developed a specialized IVM-Mix-1M dataset and introduced an innovative Discriminator Weighted Supervised Learning (DWSL) training objective, both of which are our primary contributions.
> - Especially, the DWSL training objective is very crucial for the improvements. Figure 9(a) clearly demonstrates that DWSL is the only method that can effectively leverage both the high-quality but small human data and the large but mixed-quality machine data to achieve superior results. The simple supervised training baseline, however, fails miserably in training on machine, human, or joint data.
> - Also, note that the supervised learning baseline degenerates to the previous method LISA[5] as we follow the model architectural framework established by LISA. So simply finetuning baseline models using the same dataset is not enough to get good results. DWSL, however, can enjoy considerable improvements.
>
> > **W3: The derived framework is a little bit heavy, with LLM and heavy visual rendering head involved. There might be a more efficient network framework to achieve similar performance.**
>
> - This limitation has been thoroughly discussed in Appendix A (Limitation and future work) of our paper. Exploring the training of smaller models and more direct methods to achieve comparable results remains a promising direction for future work.
> - However, it’s crucial to highlight that the IVM-7B can significantly enhance the performance of larger models like GPT-4, offering a relatively effective solution.
>
> >**Q1.1: How does this approach adapt to images with distinct image resolution?**
> - We follow the official image transformation tool in previous work[5, 6] to standardize the input image resolution: 1024x1024 for SAM and 386x386 for MLLM (Multimodal Large Language Model).
>
> >**Q1.2: Is there any specifically designed relative position embedding?**
> - No, we follow the standard position embedding from [5, 6].
>
> >**Q2: Is there any plan to extend this work to more complicated robot action...**
> - Sure! This will be very promising and interesting!
> - Also, note that although the task is short-horizon, our evaluation already poses a significant challenge for current robot models due to the huge adversarial human disturbances and visual distractors. Please see supplementary materials for those interesting videos.
>
> [1] Kosmos-2: Grounding Multimodal Large Language Models to the World, 2023
>
> [2] Lion: Empowering multimodal large language model with dual-level visual knowledge, CVPR 2024
>
> [3] Shikra: Unleashing Multimodal LLM’s Referential Dialogue Magic, 2023
>
> [4] Ferret: Refer and Ground Anything Anywhere at Any Granularity, ICLR 2024
>
> [5] LISA: Reasoning Segmentation via Large Language Model, CVPR 2024
>
> [6] Segment Anything, ICCV 2023

---

### Author Rebuttal · Authors · 2024-08-02

## **General Response**
We sincerely thank all the reviewers for the positive feedback and constructive comments on our work!
Here, we summarize the contents in the attached PDF.

1. For Reviewer EbuK (W1) & zLHv (Q4): We updated the Figure 6 (modified) according to the constructive comments.
2. For Reviewer zLHv (Q5): We added the IVM-enhanced MLLM inference pipeline in Figure a.
3. For Reviewer EbuK (Q1): We provided some examples from the simple GQA, SQA, and VQAv2 benchmarks.

---

### Decision · Program_Chairs · 2024-09-25

**Decision:**

Accept (poster)

**Comment:**

Thank you for the detailed rebuttal. After reviewing the reviewers' comments and the authors' responses, I appreciate the strong contribution of this paper to the field of multimodal instruction following. The proposed Instruction-Guided Visual Masking (IVM) model effectively enhances multimodal models by enabling precise focus on task-relevant regions, showing significant promise for improving performance across various downstream tasks. Given the strengths and the thorough validation of the proposed methods, I recommend accepting this paper.